# Alcohol Use and Its Affordability in Adolescents in Slovakia between 2010 and 2018: Girls Are Less Adherent to Policy Measures

**DOI:** 10.3390/ijerph18105047

**Published:** 2021-05-11

**Authors:** Róbert Ochaba, Tibor Baška, Martina Bašková

**Affiliations:** 1Department of Public Health, Faculty of Health Care and Social Work, Trnava University, 918 43 Trnava, Slovakia; robert.ochaba@uvzsr.sk; 2Department of Public Health, Jessenius Faculty of Medicine in Martin, Comenius University in Bratislava, 036 01 Bratislava, Slovakia; tibor.baska@uniba.sk; 3Department of Midwifery, Jessenius Faculty of Medicine in Martin, Comenius University in Bratislava, 036 01 Bratislava, Slovakia

**Keywords:** adolescents, alcohol, affordability, socioeconomic groups

## Abstract

Background: The article analyzes selected indicators of alcohol use (weekly use, drunkenness within last month) and the ability of adolescents to buy alcohol in Slovakia between 2010 and 2018. Methods: Health Behavior in School-Aged Children (HBSC) is a cross-sectional questionnaire study. A standardized uniform questionnaire was used to create a representative sample of 15-year-old adolescents. Two surveys carried out in Slovakia in 2010 (*n* = 1568) and 2018 (*n* = 1298) were analyzed. Results: Weekly alcohol use and drunkenness declined only in boys, not in girls. Affordability of alcohol (not being prevented from buying it) declined among weekly drinking boys (from 60.4 to 34.1%) but remained almost unchanged in girls from a higher socioeconomic group compared to those from a lower one (57.9% vs. 30.6% in 2018). Conclusions: Affordability of alcohol in boys decreased with a decline in alcohol use, corresponding with implemented legislative measures. However, it remained unchanged in girls from a higher socioeconomic group.

## 1. Introduction

Excessive alcohol consumption has been recognized as one of the leading preventable risk factors of premature loss of health due to either numerous health disorders or injuries leading to a considerable social burden [1]. Moreover, these consequences include effects on people other than drinkers themselves [2]. Despite a slight decline within the last few years, the alcohol-attributable social health burden still remains, particularly high in Central and Eastern Europe. However, there is high variability in the trends across European countries. The declining trend has been most pronounced in Northwestern Europe and Mediterranean countries, while in Central and Eastern Europe, it has remained almost unchanged [3]. The situation in Slovakia corresponds with that in other countries of the European Union: the consumption of pure alcohol per capita declined from 12.1 L in 1998 to 10.2 L in 2018. However, this exceeds the WHO European Region (7.8 L), and thus Slovakia, together with other countries of Central and Eastern Europe, still ranks among countries with relatively high alcohol consumption [4,5]. On the other hand, adolescent alcohol use in Slovakia declined over the last decades, particularly between 2006 and 2014, and gender differences, i.e., dominance of boys, were alleviated [6]. This corresponds with a development previously seen in Western Europe [7]. Currently, the situation regarding adolescent alcohol use in Slovakia shows patterns very similar to other European Union countries. This development indicates increasing globalizing effects, the disappearance of local specifics, and a sharing of common determinants of initiation and development of alcohol use throughout Europe.

Considering the importance of public health in relation to alcohol use, adolescents rank among the most significant population groups: This age group is crucial for the initiation and development of use, determining further drinking patterns as well as the epidemiological situation in the future. Regular alcohol use during adolescence is associated with various kinds of risky behavior such as unsafe sex, injuries, and use of other addictive substances. Besides detrimental health effects pronounced in this age group, early initiation of regular drinking increases the risk of excessive and problem drinking at later age and adulthood [8,9]. Binge drinking leading to drunkenness, in particular, presents a special issue [10].

Socioeconomic (SE) status is one of the factors determining alcohol use in adolescents. Studies focused on this issue present ambiguous results. Some studies indicate a higher probability of alcohol use in adolescents from higher SE groups, as seen in Denmark [11], the Baltic countries [12], and northwest England [13]. On the other hand, a different situation was observed in Finland, where girls with low perceived family wealth experienced drunkenness more frequently, and no remarkable SE differences were found among boys [14]. The heterogeneity of above-mentioned findings can be explained by the effect of differences in geographical–cultural background, and thus different situations are observed across countries [15]. When combining data from 34 countries, heterogeneity across countries indicated an insignificant association between SE status and alcohol use [16].

Control measures focused on adolescence play a crucial role in preventing the overall health and social impact of excessive alcohol use in a society. Among them, policies aimed at availability (i.e., ban of sale to minors) and pricing have proved to be effective [8,17]. However, their impact is significantly mediated by an overall social environment, namely prevalence of drinking among the general adult population [10] together with culturally/historically based drinking patterns [18], family influences, and peer pressure [8,19]. These underlying factors can explain the relatively weak effect of preventive measures shown in several studies [20,21].

In Slovakia, within the relevant time period, two amendments of the existing Act No. 219/1996 Coll. on Protection against Alcohol Abuse came into force, having a possible significant effect on underage access to alcohol. Firstly (Act. 214/2009 Coll.), the presence of persons up to 15 years old has been banned in public places that serve alcoholic beverages after 9:00 p.m., unless accompanied by their legitimate representatives (i.e., parents, legal guardians, etc.). Secondly (Act. 88/2013 Coll.), local municipalities have the option to take into consideration local situation and to specify additional public places where the sale or serving of alcoholic beverages should be banned, i.e., places beyond ones already defined in the legislation at the national level. Moreover, on 3 July 2013, the Government of the Slovak Republic approved an official alcohol control policy based on a strategy of the World Health Organization [22], considering it one of the main priorities of public health. These activities resulted in allowing for the enforcement of existing norms and informational activities to decrease social tolerance toward youth alcohol consumption.

However, most of the studies examining the impact of an alcohol control policy did not analyze gender or SE differences and considered the issue in general [17,18,21,23]. Therefore, little is known about how sociodemographic determinants of adolescents such as their gender and SE family background can influence their adherence to control measures and subsequent changes in their drinking behavior.

The aim of the study is to analyze differences between boys and girls as well as between higher and lower SE groups, considering changes in regular (weekly) alcohol consumption, high-risk drinking patterns (indicated as previous month’s drunkenness), and affordability of alcohol (indicated as a perception of ability to buy alcohol among weekly drinkers) from 2010 to 2018. The study uses Health Behavior in School-Aged Children (HBSC) data.

The data on the epidemiological situation of alcohol consumption in adolescents can contribute to an understanding of the effect of legislative measures, considering gender and SE situation, and to identifying subpopulations with lower adherence to alcohol control policy.

## 2. Materials and Methods

Health Behavior in School-Aged Children is an international, school-based cross-sectional study. Its standardized design makes it possible to create harmonized datasets appropriate for cross-country comparisons as well as for identifying changes over time. In Slovakia, four HBSC surveys have been carried out: in school years 2005/2006, 2009/2010, 2013/2014, and 2017/2018, i.e., in May–June 2006, 2010, 2014, and 2018.

Data were collected via uniform anonymous questionnaires completed in classrooms at schools. The questionnaires include mandatory modules of questions used in every participating country as well as optional ones containing sets of questions based on the special needs of individual countries. In surveys carried out in 2010 and 2018, a set of questions focusing on the availability of psychoactive substances were included in the optional modules.

The samples were compiled in accordance with the structure of the educational system in Slovakia and stratified by region and type of school in order to obtain representative data on 11-, 13-, and 15-year-old adolescents. Two-step sampling was used, following the standardized research protocol [24]. In the first step, participating schools were randomly selected with probability proportional to size using an official list of all schools obtained from the Slovak Institute of Information and Prognosis for Education. The sample of schools was stratified by region (eight self-governing administrative regions) and type of school (elementary schools comprising the 1st–9th grades, and eight-year grammar schools comprising the 6th–13th grades). In the second step, within the participating schools, classes were randomly selected to collect questionnaire data. This sampling approach provides representative data on the national level reflecting the actual epidemiological situation in the country.

This study was approved by the Ethics Committee of the Faculty of Medicine at Pavol Jozef Šafárik University in Kosice in 2009 (No. 82/2009) and 2017 (No. 16N/2017). Parents were informed about the study via the school administration and, using a written informed consent form, could opt out if they disagreed with their child’s participation. Participation in the study was fully voluntary and anonymous, with no explicit incentives provided for participation. The standardized research protocol and uniform internationally used questionnaire provides data comparable across countries as well as making possible comparisons across time. Further information on HBSC surveys in Slovakia can be found in our previous article [6]. The authors of this article are members of the multidisciplinary HBSC Slovakia investigation team and were participating in the preparation of protocol, arrangement of fieldwork as well as data evaluation.

This study analyzes HBSC data from Slovakia from the surveys carried out in 2010 and 2018 on weekly alcohol consumption and drunkenness within the last 30 days, as well as the ability to buy alcohol considering SE family background. Since the rates in 11- and 13-year-old respondents were very low, we included only 15-year-old respondents to make the analysis distinct and clear. The analyzed samples included 1568 respondents (771 boys) in 2010 and 1293 respondents (703 boys) in 2018. Response rates were 79.5% and 60.0%, respectively. Dropouts were caused mostly by the absence of children due to illness, other personal reasons, and refusal to be involved in the study. Therefore, they were unrelated to the analyzed variables and could not significantly bias the results.

Weekly alcohol consumption was measured by the question “At present, how often do you drink anything alcoholic, such as beer, wine, or spirits?” The following kinds of beverages were stated: Beer, wine, spirits, alcopops, and other drinks. In each kind, possible responses were “every day”, “every week”, “every month”, “sometimes”, and “never”. An answer of at least “every day” or “every week” in at least one of the presented beverages was considered weekly drinking.

Drunkenness within the last month was measured by the question “On how many days (if any) have you got drunk during the last 30 days?” Possible responses were “never”, “1–2 days”, “3–5 days”, “6–9 days”, “10–19 days”, “20–29 days”, and “30 days or more”. All answers except “never” were considered positive.

Ability to buy alcohol was measured by the question “When you wanted to buy alcohol in a shop, bar, discount store, etc., did anyone refuse to sell it to you because of your age?” Possible responses were “I did not buy alcohol”, “Yes, someone refused to sell it to me”, and “No, I bought it”. The answer “No, I bought it” was considered positive. This variable was analyzed only among respondents reporting weekly alcohol consumption.

SE background was measured by the Family Affluence Scale (FAS), which consisted originally of four questions [25]: “Does your family own a car, van, or truck” (No = 0, Yes, one = 1, Yes, two or more = 2), “Do you have your own bedroom for yourself?” (No = 0, Yes = 1), “How many computers does your family own?” (None = 0, One = 1, Two = 2, More than two = 3), “How many times did you and your family travel out of Slovakia for a holiday/vacation last year?” (Not at all = 0, Once = 1, Twice = 2, More than twice = 3). To take into account changes in family consumption patterns, two more questions were added to this instrument [26] in the last survey: “How many bathrooms (room with a bath/shower or both) are in your home?” (None = 0, One = 1, Two = 2, More than two = 3), “Does your family have a dishwasher at home?” (No = 0, Yes = 1). Responses were scored and summed to form the final score. Values up to median were considered as a lower SE group (subpopulation) and above median as a higher one.

The results are presented as absolute numbers and respective percentages. Differences between rates were statistically analyzed using a chi-square test. As a level of statistical significance of the difference, *p* < 0.05 was used.

## 3. Results

Table 1 summarizes all the results. It presents prevalence rates of analyzed variables in boys and girls as well as in lower and higher socioeconomic groups.

### 3.1. Changes between 2010 and 2018

During the studied period, prevalence rates of weekly alcohol drinkers declined particularly among boys (from 28.8 to 20.3%), holding statistical significance both in lower and higher SE groups. In girls, the decline was less pronounced (from 15.5 to 10.4%) and lost its statistical significance when split into SE groups (Figure 1).

The decline of respondents reporting being drunk at least once a month was significant only in boys (Figure 1), particularly in the lower SE group (from 25.5 to 16.0%). In girls, the situation did not change, and the prevalence rate varied between 18.9% (lower SE group in 2010) and 14.8% (higher SE group in 2018).

Changes in the affordability of alcohol between 2009/2010 and 2017/2018 showed a different picture in boys and girls. While the proportion of boys not being prevented from buying alcohol sharply declined, particularly in lower SE group (from 60.4 to 30.9%), the situation in girls did not change significantly (Figure 1).

### 3.2. Differences between Boys and Girls

Weekly alcohol consumption remarkably dominated in boys regardless of the SE group, particularly in 2010 (Figure 2). However, considering positive reports on drunkenness, boys prevailed over girls only within the lower SE group in 2010 (24.4% vs. 18.9%), while in other subgroups, the gender differences were insignificant. The prevalence of weekly drinkers not prevented from buying alcohol was only insignificantly different between genders, except the lower SE group in 2009/2010, where boys dominated (60.4% vs. 38.0%).

### 3.3. Drinking between Lower and Higher SE Groups

Differences between SE groups were insignificant in all measured indicators and subgroups except girls in 2017/2018 (Figure 3), where respondents from a higher SE group more frequently reported not being prevented from buying alcohol (57.9% vs. 30.6%).

## 4. Discussion

Alcohol use remains widespread in 15-year-old adolescents despite a remarkable decline in the last couple of years. However, this decline refers mainly to boys, which challenges expectations of traditional gender differences. Moreover, the decline in drunkenness was not so steep and was seen only among boys from the lower SE group. It indicates a change in the structure of current drinkers, i.e., a relative increasing proportion of high-risk drinkers.

Changes in affordability refer particularly to boys and the decline of drinkers not prevented from buying alcohol was remarkable in both lower and higher SE groups. On the other hand, the situation for girls is quite different. No remarkable decline was seen, i.e., the affordability did not change. However, it seems that the SE situation plays a more important role for girls than boys, since the predominance of larger affordability was among girls from the higher SE group.

We should positively consider the decline of alcohol use [5]. However, the results indicate that the previously seen predominance of males gets to disappear [27]. It means that specific features of women’s habits will fade away, including a traditionally lower occurrence of binge drinking [28]. The relative increase of drunkenness among regular drinkers should be a reason for concern.

Between 2010 and 2013, significant legislative changes regarding alcohol control came into force in Slovakia. They are focused mostly on restricting underage affordability of alcohol. Reduced ability to buy alcohol can be considered one of the reasons for the positive development of the epidemiological situation in Slovakia, i.e., the decline of regular weekly alcohol use among adolescents. A comparison with our previous study analyzing trends of alcohol use in Slovakia [6] supports the possible role of the above-mentioned legislative changes. If compared with the latest results in 2018, weekly drinking almost did not change after 2014 in both boys and girls (from 21.0 to 20.4% and from 11.9 to 10.4%, respectively). It indicates that the most significant decline occurred before 2014, corresponding with the time when the policy changes came into force.

However, our results indicate widespread affordability despite age restrictions, consistent with findings in other European countries [29,30,31,32], indicating that the situation is far from ideal. Analyzing the results of HBSC Slovakia in 2009/2010 and 2017/2018, it seems that the impact of a changing social environment is not consistent across all population subgroups, and a different picture can be seen in boys and girls. Unlike several studies showing a higher level of affordability in girls [29,33], in our analysis, boys reported being more successful in buying alcohol in 2010 (lower SE group). However, this situation changed within the given period: While affordability in boys remarkably declined regardless of SE group, it remained almost unchanged in girls. Gender differences were not observed in 2018, and the situation is becoming similar to that in Western Europe. Moreover, the results indicate a relative increase in affordability in girls from a higher SE group, resulting in a significant difference between SE groups (57.9% vs. 30.6%). Higher affordability of alcohol in girls, particularly those belonging to a higher SE group, can be explained by two possible behavioral traits: (1) They can better mask their underage appearance, looking and behaving more mature. Thus, these girls are more successful in purchasing alcohol. (2) They more frequently employ the assistance of older friends to purchase alcohol [34]. Although most studies dealing with these aspects did not analyze gender or SE differences, some of them support this view, as they demonstrated a higher success rate in buying alcohol among girls [29,33,35].

Our findings suggest that to understand the role of social determinants, namely, legislation and policy, detailed analyses within specific subpopulations are needed and not just to study a target population as a whole. Attention should be paid to further development, i.e., differences across the SE subpopulation and between boys and girls: Which factor plays a role in differences in affordability, or why are some subpopulations more resistant to policy measures than others? In 2018, another significant amendment (Act 163/2018 Coll.) was passed. This norm explicitly bans drinking under the age of 18 (the previous norms only banned the underage sale or serving of alcohol) and makes underage access to alcohol even more difficult. Thus, further investigation of the issue is important, making it possible to identify subsequent changes across sociodemographic population groups.

Considering the possible limitations, we should mention that the results are based on self-reports, thus, the prevalence rates we found could differ somewhat from the actual situation. However, as standardized uniform methods were used, the validity and reliability of the results are the same across the surveys. Thus, the established differences should be considered valid, reflecting actual changes. The used data originate from cross-sectional studies, and thus, neither correlations nor causal associations can be analyzed. On the other hand, evaluation of differences in prevalence rates reflects changes in the epidemiological situation and thus meet the given goals. We are aware that the study concerns a relatively narrow age group and thus cannot reflect changes during adolescence. Thus, we should carefully interpret our findings considering other age groups. After all, the chosen age group (15 years) is particularly significant for the development of alcohol use and thus the results have contributive practical implications. SE status was measured via Family Affluence Scale (FAS). To avoid immoderate disintegration of the studied sample, we divided it only into two subgroups (higher vs. lower) using a median as a cut-off. We are aware that such an approach cannot exactly reflect SE differences and would not be appropriate for deeper analysis. However, it meets a need to define the subpopulation of different SE backgrounds to identify differences in alcohol use and related factors.

## 5. Conclusions

Between 2010 and 2018, social affordability of alcohol declined only in boys, not in girls. Moreover, access of alcohol among adolescents, despite positive development in boys, remains a significant issue. It applies to girls in a higher SE group, which showed the lowest adherence to measures focused on access to buying alcohol. Therefore, our findings outline implications for further research, which should be focused on causes of gender and SE differences in adherence to policy measures. It could contribute to understanding factors making preventive interventions more effective.

## Figures and Tables

**Figure 1 ijerph-18-05047-f001:**
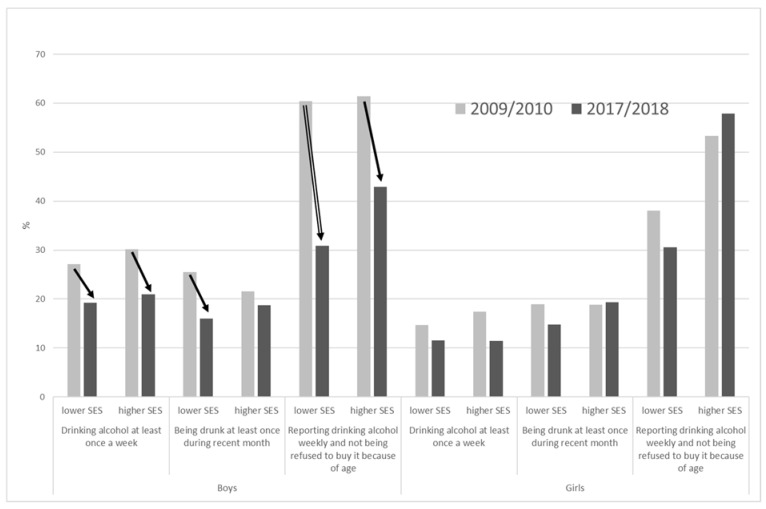
Changes in prevalence rate of weekly alcohol use, drunkenness, and ability to buy alcohol in 15-year-old boys and girls between 2010 and 2018. *↘*—significant decline (*p* < 0.05). ⇘—significant decline (*p* < 0.001).

**Figure 2 ijerph-18-05047-f002:**
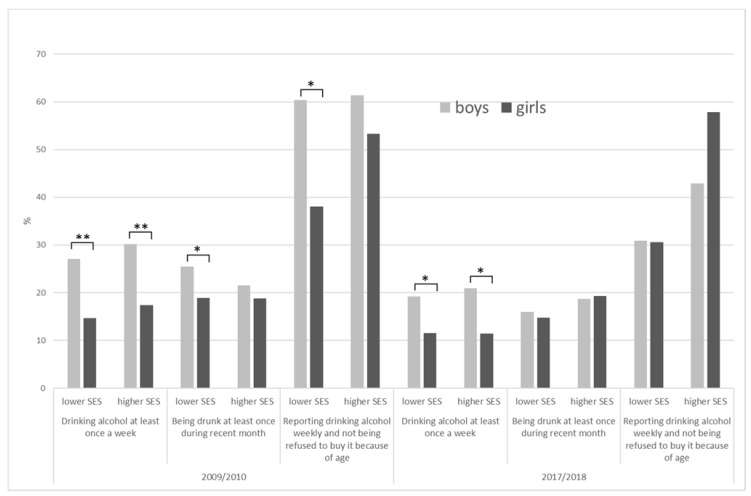
Prevalence rate of weekly alcohol use, drunkenness, and ability to buy alcohol in 15-year-old adolescents by SE groups—differences between boys and girls. *—significant difference (*p* < 0.05). **—significant difference (*p* < 0.001).

**Figure 3 ijerph-18-05047-f003:**
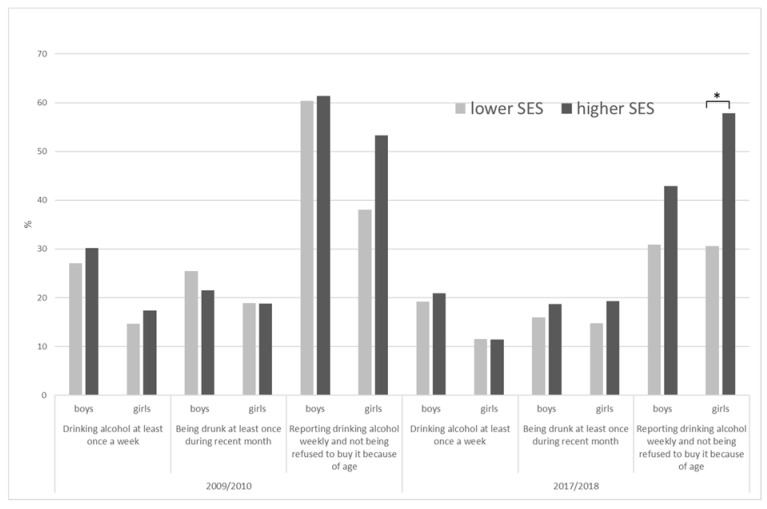
Prevalence rate of weekly alcohol use, drunkenness and availability to buy alcohol in 15 years old boys and girls—differences between lower and higher SE group. *—significant difference (*p* < 0.05).

**Table 1 ijerph-18-05047-t001:** Prevalence of positive reports in 15-year-old adolescents by gender and socioeconomic status (absolute numbers and percentages).

Variable	Years		Boys		Girls	
Drinking alcohol at least once a week	2009/2010	All	225	(28.7%)	123	(15.6%)
		lower SE status	112	(27.1%)	71	(14.7%)
		higher SE status	83	(30.2%)	45	(17.4%)
	2017/2018	all	129	(20.4%)	58	(10.4%)
		lower SE status	56	(19.2%)	37	(11.6%)
		higher SE status	42	(21.0%)	19	(11.5%)
Being drunk at least once during last month	2009/2010	all	193	(24.4%)	152	(18.9%)
		lower SE status	107	(25.5%)	93	(18.9%)
		higher SE status	60	(21.6%)	49	(18.8%)
	2017/2018	all	106	(17.4%)	84	(15.5%)
		lower SE status	45	(16.0%)	46	(14.8%)
		higher SE status	36	(18.7%)	31	(19.3%)
Reporting drinking alcohol weekly and not being prevented from buying it because of age	2009/2010	all	136	(60.4%)	55	(44.7%)
		lower SE status	67	(60.4%)	27	(38.0%)
		higher SE status	51	(61.4%)	24	(53.3%)
	2017/2018	all	44	(34.1%)	23	(39.7%)
		lower SE status	17	(30.9%)	11	(30.6%)
		higher SE status	18	(42.9%)	11	(57.9%)

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
