# Peer review of "Alcohol Use and Its Affordability in Adolescents in Slovakia between 2010 and 2018: Girls Are Less Adherent to Policy Measures"

_ijerph, 2021, doi:10.3390/ijerph18105047_

Round 1
Reviewer 1 Report
Page 1: English language issues in a few places
Page 2: English language issues in places. Wrong tense or wrong word. Line 80 has an incomplete sentence and awkward wording for the full paragraph.
Page 3-6: English language issue in places. Line 200+: wording is awkward
Outside of the English language edits needed, the Figure 1 chart shows what I believe is an incorrect posting of the results for SE for the last question on reporting of drinking and not being refused.
I suggest also that mention be made more clearly that this is a limited study. First, the charts showed only 15 year olds' data but the surveys went to 11, 13 and 15 year olds. There was no mention why only these age groups and not all teens. Also, it is hard to generalize from just the 15 year olds anything anything from looking only at 15year olds. Can you explain why you made that decision. I say this because, as a youth worker, there is a big difference in terms of decisions made about key issues within that age range. How a 15 year things is different from that of a much younger 11 year old. I would rather see all of the data and not just one age group. After all, a lot happens in the life of an 11 year old compared to a 15 year old.
More should be said about the fact that there was not a great deal of difference between girls and boys in reality - emotionally, physically, cognitively, spiritually.
I also would like a little picture painted of Slovakia to see how that impacts the teens' decisions in terms of alcohol. Are they depressed because of the environment? Why do they drink? Is it because mom and dad regularly drink something - like a glass of wine or a bottle of beer? My experience is that, as a rule, Europeans tend to drink alcohol at meals and that often their older children are allowed to drink with them.
Also nothing was said about the power of peers and peer pressure, yet we know that this is important for teens. Also, we don't know if drinking and alcoholism is a serious issue in Slovakia compared with other countries and compared with substance abuse. The authors note briefly the Slovakian laws limiting drinking. However, as seen in the US, for example, there are ways around the laws. So is this an issue? At one point, the authors say, drinking is common. How common?
I feel that there are assumptions made that the reader is aware of all of the issues. Then statements are made outright without explanation or any background provided. I would just like a little more background and description.
Certainly this is a good topic and issue to address. As an educator of youth, I am concerned about alcohol consumption among teens. I just would like to see some edits made in the paper to sharpen these points.
Author Response
Response for Reviewer 1
Page 1: English language issues in a few places
Response: The manuscript has been edited by the experienced native English speaker (provided via English language editing by MDPI).
Page 2: English language issues in places. Wrong tense or wrong word. Line 80 has an incomplete sentence and awkward wording for the full paragraph.
Response: The whole respective paragraph was completely rephrased.
Page 3-6: English language issue in places. Line 200+: wording is awkward
Response: The respective sentences were rephrased and edited..
Outside of the English language edits needed, the Figure 1 chart shows what I believe is an incorrect posting of the results for SE for the last question on reporting of drinking and not being refused.
Response : We checked the respective figure very carefully, however, it is correct. We corrected special signs (arrow and double arrow) in the caption.
I suggest also that mention be made more clearly that this is a limited study. First, the charts showed only 15 year olds' data but the surveys went to 11, 13 and 15 year olds. There was no mention why only these age groups and not all teens. Also, it is hard to generalize from just the 15 year olds anything anything from looking only at 15year olds. Can you explain why you made that decision. I say this because, as a youth worker, there is a big difference in terms of decisions made about key issues within that age range. How a 15 year things is different from that of a much younger 11 year old. I would rather see all of the data and not just one age group. After all, a lot happens in the life of an 11 year old compared to a 15 year old.
Response: We agree with your valuable comments. In fact, it is rather narrow age group to provide a complex picture on epidemiological situation in adolescents. However, we decided to include only 15 years old respondents because among younger age groups the found rates were too low to provide material for reasonable analysis. We added an explanation into Materials and methods as well as we mentioned this issue in Discussion among limitations.
More should be said about the fact that there was not a great deal of difference between girls and boys in reality - emotionally, physically, cognitively, spiritually.
Response : Really, it is a very interesting and important issue. We mentioned this in Introduction while describing overall situation in adolescents as a possible result of increasing globalizing effects, the disappearance of local specifics, and a sharing of common determinants of initiation and development of alcohol use throughout Europe.
I also would like a little picture painted of Slovakia to see how that impacts the teens' decisions in terms of alcohol. Are they depressed because of the environment? Why do they drink? Is it because mom and dad regularly drink something - like a glass of wine or a bottle of beer? My experience is that, as a rule, Europeans tend to drink alcohol at meals and that often their older children are allowed to drink with them.
Response: We added more information on epidemiological situation in Slovakia in the European context. However, we think that more detailed considerations on this issue would be beyond the topic and would disproportionally extent the text of the manuscript.
Also nothing was said about the power of peers and peer pressure, yet we know that this is important for teens. Also, we don't know if drinking and alcoholism is a serious issue in Slovakia compared with other countries and compared with substance abuse. The authors note briefly the Slovakian laws limiting drinking. However, as seen in the US, for example, there are ways around the laws. So is this an issue? At one point, the authors say, drinking is common. How common?
Response: Many thanks for this important comment. We mentioned it in Introduction: „...However, their impact is significantly mediated by an overall social environment, namely prevalence of drinking among the general adult population [10] together with culturally/historically based drinking patterns [17], family influences, and peer pressure [7,18]. These underlying factors can explain the relatively weak effect of preventive measures shown in several studies [19,20].
I feel that there are assumptions made that the reader is aware of all of the issues. Then statements are made outright without explanation or any background provided. I would just like a little more background and description.
Response : We extended background information on epidemiological situation both in adult and adolescent population in Introduction.
Certainly this is a good topic and issue to address. As an educator of youth, I am concerned about alcohol consumption among teens. I just would like to see some edits made in the paper to sharpen these points.
Response : Thank you very much for this supportive and stimulating comment.
Reviewer 2 Report
The authors reported results of the HBSC 2010 and 2018 on drinking behaviors of adolescents. Interesting differences were found between boys and girls across various SE categories.
However, the manuscript was not very easy to read because of the language. The authors might want to rephrase some of the sentences to make them more clear in meanings.
Secondly, the study was conduced based on some valuable data, however the statistical analysis could be more in-depth. The current manuscript was too much like a survey report instead of a scientific investigation. The authors might have to think of their study constructs, hypotheses, and ways of present and exam them.
Author Response
Response for Reviewer 2
The authors reported results of the HBSC 2010 and 2018 on drinking behaviours of adolescents. Interesting differences were found between boys and girls across various SE categories.
Response : Many thanks for this positive comment. We highly appreciate it.
However, the manuscript was not very easy to read because of the language. The authors might want to rephrase some of the sentences to make them more clear in meanings.
Response: The manuscript has been edited by the experienced native English speaker (provided via English language editing by MDPI).
Secondly, the study was conduced based on some valuable data, however the statistical analysis could be more in-depth. The current manuscript was too much like a survey report instead of a scientific investigation. The authors might have to think of their study constructs, hypotheses, and ways of present and exam them.
Response: We are aware of it. However, a cross-sectional design of the HBSC study does not allow to make analyses of causal associations. On the other hand, the given approach is appropriate considering the aims, i.e. to analyze changes of prevalence rates in special population groups.
We completely rephrased the paragraph on aims to better express our intentions: “…The aim of the study is to analyze differences between boys and girls as well as be-tween higher and lower SE groups, considering changes of regular (weekly) alcohol consumption, high-risk drinking patterns (indicated as previous month’s drunken-ness), and affordability of alcohol (indicated as a perception of ability to buy alcohol among weekly drinkers) from 2010 to 2018. The study uses Health Behaviour in School-Aged Children (HBSC) data….”
Reviewer 3 Report
Although the key information is present in the abstract, please review for consistent tense and check grammar. Extensive grammar revision is necessary throughout the paper. Please provide a statement of third-party approval that you secured to conduct this study (e.g., Institutional Review Board for the Protection of Human Subjects) or if your local context does not require such oversight, then please indicate this and describe how you ensured ethical research practice to protect participants’ safety, privacy, and confidentiality. If the study was deemed to be exempted or excluded from IRB review, please make note of it.
Provide information of approval by the HBSC International Coordinating Centre. Also, include more detail on the target population selected. Clarify the role of the authors in data collection. Please add an about the researcher(s) section and your connection to this study. How does this align with personal interests, professional work, etc., to help the reader place you directly in the center of your work?
Author Response
Response for Reviewer 3
Although the key information is present in the abstract, please review for consistent tense and check grammar. Extensive grammar revision is necessary throughout the paper. Please provide a statement of third-party approval that you secured to conduct this study (e.g., Institutional Review Board for the Protection of Human Subjects) or if your local context does not require such oversight, then please indicate this and describe how you ensured ethical research practice to protect participants’ safety, privacy, and confidentiality. If the study was deemed to be exempted or excluded from IRB review, please make note of it.
Response: The manuscript has been edited by the experienced native English speaker (provided via English language editing by MDPI).
We agree with this meaningful comment. Indeed, this information should be included into the text. IRB review was not required, but we added an information on ethical committee approval in Materials and Methods:”… This study was approved by the Ethics Committee of the Faculty of Medicine at Pavol Jozef Šafárik University in Kosice in 2009 (No. 82/2009) and 2017 (No. 16N/2017). Parents were informed about the study via the school administration and, using a written informed consent form, could opt out if they disagreed with their child’s participation. Participation in the study was fully voluntary and anonymous, with no explicit incentives provided for participation…”.
Provide information of approval by the HBSC International Coordinating Centre. Also, include more detail on the target population selected. Clarify the role of the authors in data collection. Please add an about the researcher(s) section and your connection to this study. How does this align with personal interests, professional work, etc., to help the reader place you directly in the center of your work?
Response: Since we used only national data, approval by the HBSC International Coordinating Centre was not needed. Authors of the manuscript are members of the national HBSC team and are affiliated as university teachers. They were involved into development of research protocol and actively participated in implementation of the study as well as data processing.
Round 2
Reviewer 1 Report
The edits made the article more clear and improved it overall. I have just a few small edits:
Line 51: The end of the sentence – “. . . as well as future society” does not make sense.
Line 57: This paragraph needs strengthening when you talk about gender. There is only one comment made concerning girls with lower family wealth. Then the next sentence talks about how heterogeneity can be explained. However, nothing was said about boys. So this comments seems to just hang there without connecting anywhere.
Line 94: “. . . adolescents such as gender. . .”
Line 249: The sentence beginning “However, the results. . .” needs to be clearer. One has to read through it several times to get the meaning of it. Also its does not seem to kit with the sentence right after it. So this section can be smoother.
Just one comment more: I personally like to see more of a concluding section that offers suggestions for further research (as you seemed to do) and also some comments about possible learnings or take aways for the reader. But it may be that the publisher does not wish to see such conclusions or it may be the difference between US thinking and European thinking.
Author Response
The edits made the article more clear and improved it overall. I have just a few small edits:
Many thanks for your encouraging evaluation!
Line 51: The end of the sentence – “. . . as well as future society” does not make sense.
Yes, we agree. We rephrased it: “epidemiological situation in the future.”
Line 57: This paragraph needs strengthening when you talk about gender. There is only one comment made concerning girls with lower family wealth. Then the next sentence talks about how heterogeneity can be explained. However, nothing was said about boys. So this comments seems to just hang there without connecting anywhere.
We added the respective information: “On the other hand, a different situation was observed in Finland, where girls with low perceived family wealth experienced drunkenness more frequently and no remarkable SE differences were found among boys [14]. The heterogeneity of above mentioned findings can be explained….”
Line 94: “. . . adolescents such as gender. . .”
We rephrased it:”… of adolescents such us their gender...“
Line 249: The sentence beginning “However, the results. . .” needs to be clearer. One has to read through it several times to get the meaning of it. Also its does not seem to kit with the sentence right after it. So this section can be smoother.
Yes, we agree, the formulation is rather clumsy. We rephrased it: “However, the results indicate that a previously seen predominance of males gets to disappear.“
Just one comment more: I personally like to see more of a concluding section that offers suggestions for further research (as you seemed to do) and also some comments about possible learnings or take aways for the reader. But it may be that the publisher does not wish to see such conclusions or it may be the difference between US thinking and European thinking.
We modified the last sentence of the Conclusions: “Therefore, our findings outline implications for further research which should be focused on causes of gender and SE differences in adherence to policy measures. It could contribute to understanding of factors making preventive interventions more effective.
Reviewer 2 Report
I appreciate the effort of authors on revising the manuscript. The quality of language significantly improved after English editing.
However, the other main concern I had, the depth of the statistical analysis, was still insufficient, which consequently limited the novelty, interest, and scientific soundness of this study.
Cross-sectional investigations are not able to conclude the directions of associations or causal relationships. However, there are still many methods, such as correlation or regression, which can help us to get more information from the data collected.
I suggest the research team review again on the hypothesis and aims of the study, apply more statistical methods, and revise the manuscript to be come more informative for the readers. Thank you.
Author Response
I appreciate the effort of authors on revising the manuscript. The quality of language significantly improved after English editing.
However, the other main concern I had, the depth of the statistical analysis, was still insufficient, which consequently limited the novelty, interest, and scientific soundness of this study.
Cross-sectional investigations are not able to conclude the directions of associations or causal relationships. However, there are still many methods, such as correlation or regression, which can help us to get more information from the data collected.
I suggest the research team review again on the hypothesis and aims of the study, apply more statistical methods, and revise the manuscript to be come more informative for the readers. Thank you.
The study was focused to evaluate changes of epidemiological data on alcohol use, drunkenness and alcohol affordability over time. Since two cross-sectional studies were compared and differences of respective prevalence rates were analyzed. We think that adequate methods were used to meet this goal. Anyway, we added this issue into the Limitations: “The used data originate from cross-sectional studies and thus neither correlations nor causal associations can be analyzed. On the other hand, evaluation of differences in prevalence rates reflect changes of epidemiological situation and thus meet the given goals.”
Reviewer 3 Report
Please consider adding an about the researcher section so you tell us more about you as the researcher(s) and your connection to this study. How does this align with personal interests, professional work, etc., to help the reader place you directly in the center of your work?
Provide brief information on the background and intent of the survey, generalizability, and reliability and validity data rather than referring the reader to another article.
Author Response
Please consider adding an about the researcher section so you tell us more about you as the researcher(s) and your connection to this study. How does this align with personal interests, professional work, etc., to help the reader place you directly in the center of your work?
We added the information on authors´ role in HBSC into the Material and Methods.
Provide brief information on the background and intent of the survey, generalizability, and reliability and validity data rather than referring the reader to another article.
We added the respective information into the Material and Methods.